# Effect of Heat Treatment Temperature on Microstructure and Properties of PM Borated Stainless Steel Prepared by Hot Isostatic Pressing

**DOI:** 10.3390/ma14164646

**Published:** 2021-08-18

**Authors:** Yanbin Pei, Xuanhui Qu, Qilu Ge, Tiejun Wang

**Affiliations:** 1Graduate School, Central Iron & Steel Research Institute, Beijing 100081, China; peiyanbin@atmcn.com; 2Institute for Advanced Materials and Technology, University of Science and Technology Beijing, Beijing 100083, China; 3Technology Department, Antai-Heyuan Nuclear Energy Technology & Materials Co., Beijing 100094, China; 4Technology Centre, Advanced Technology & Materials Co., Ltd., Beijing 100081, China; wangtj@atmcn.com

**Keywords:** borated stainless steel, heat treatment, powder metallurgy, hot isostatic pressing, field-emission scanning electron microscopy with energy-dispersive spectroscopy, microstructure, strength and plasticity

## Abstract

Borated stainless steel (BSS) with a boron content of 1.86% was prepared by a powder metallurgy process incorporating atomization and hot isostatic pressing. After solution quenching at 900–1200 °C, the phase composition of the alloy was studied by quantitative X-ray diffraction phase analysis. The microstructure, fracture morphology, and distributions of boron, chromium, and iron in grains of the alloy were analyzed by field-emission scanning electron microscopy with secondary electron and energy-dispersive spectroscopy. After the coupons were heat treated at different temperatures ranging from 900 to 1200 °C, the strength and plasticity were tested, and the fracture surfaces were analyzed. Undergoing heat treatment at different temperatures, the phases of the alloy were austenite and Fe_1.1_Cr_0.9_B_0.9_ phase. Since the diffusion coefficients of Cr, Fe, and B varied at different temperatures, the distribution of elements in the alloy was not uniform. The alloy with good strength and plasticity can be obtained when the heat treatment temperature of alloy ranged from 1000 to 1150 °C while the tensile strength was about 800 MPa, with the elongation standing about 20%.

## 1. Introduction

Borated stainless steels (BSS) are widely used in spent fuel storage racks, baskets for spent fuel storage, transportation casks, neutron shielding plates, and reactor control rods in the nuclear power industry owing to their excellent thermal neutron absorption capacity, mechanical properties, and corrosion resistance [1]. Highly borated steel, which contains more than 0.1% B, can be used as a neutron absorber. According to ASTM A887, these alloys are categorized into seven types according to boron content of 0.2–2.25%; each type is further divided into grades A and B, according to mechanical properties. BSS is generally manufactured by casting [2,3,4] or powder metallurgy (PM) [5,6,7]. Alloy manufactured by PM has a more uniform boron distribution than that of cast alloy, so grade A BSSs with higher mechanical properties are prepared by this technique.

It is generally accepted that boron solubility in ferrite and austenite is only 0.0005–0.008 at % [8]. Recent research shows that the addition of other metals, such as Cr, Mo, and V, can increase the solubility of boron to 0.185–0.515% in the cast condition and to 0.015–0.0589% in the solution-treated condition [9]. Solute boron that segregates to the austenite grain boundary delays the nucleation of ferrite on the grain boundary, thus improving the hardenability. The morphology and quantity of austenite grain boundaries change with the chemical composition and heat-treatment conditions. The boron concentration profile around austenite grain boundaries reportedly changed with cooling rate from the solid-solution temperature [1223 K]: the amount of segregation, which was estimated by the Gibbsian interfacial excess, increased when the cooling rate decreased [10].

Application of the Fe–Cr–B ternary system in the phase equilibrium of iron (steel) matrix composites and chromium-containing steels has been extensively studied. Bondar summarized the phase relationship, structure, and thermodynamics of the B–Cr–Fe system for studies published from 1958 to 2005 [11]. The mutual solubility of CrB and FeB is large, reaching compositions of (Cr_0.6_Fe_0.4_)B and (Cr_0.8_Fe_0.2_)B, respectively. Fe solubility in Cr_3_B_4_ and CrB_2_ is about 6 at % at 1100 °C and estimated at about 2 at % Fe for CrB_4_. At 1100 °C, the alternative equilibrium (αFe) + Fe_2_B occurs. Raghavan hypothesized that the solid-phase invariant reaction (γFe) + Cr_2_B ⇌ (αFe) + Fe_2_B takes place at ≈1150 °C.

The partial segregation of elements occurs by diffusion. Zhang et al. studied the solution and diffusion mechanism of boron in face-centered cubic Fe using first-principle calculations and a five-frequency model [12]. Busby et al. used two methods to determine the diffusion coefficient of boron in austenite at 950 to 1300 °C [8]. Campos et al. studied paste boriding of steel and measured boron mobility on AISI 1045 and M2 steels using the mass-balance equation, assuming that the concentration at the interface was linearly distributed [13]. Miyamoto et al. calculated that the diffusion distance at a quenching temperature of 1200 °C was 7.2 μm, which was about five times that at 900 °C. Therefore, more excess vacancies with longer diffusion distance should enhance non-equilibrium segregation of B during quenching from 1200 °C compared with quenching from 900 °C [14].

PM is the first choice for preparing high-performance and high-boron stainless steels. Stinner discussed the application and properties of BSS prepared by PM, with emphasis on ATI 304B7™ alloy, in which the homogenous microstructure leads to high ductility and toughness, improved neutron absorption, and better corrosion resistance [6]. Zhou et al. investigated the effect of borides on the hot-deformation behavior and microstructure evolution of PM high-borated stainless steel [5]. Hot compression tests at temperatures of 950–1150 °C and strain rates of 0.01–10 s^−1^ were performed. Francisco et al. focused on the role of boron as a sintering additive in PM of AISI 316L austenitic stainless steel and investigated the effect of boron regarding densification, microstructure, hardness, and micro-scale abrasion [7]. Park et al. investigated the effects of boron on densification for sintering via master sintering curves and microstructure analysis based on image processing using mixtures of different boron contents in 316L stainless steel powder [15]. The distributions of boron phases and elemental boron have a strong influence on the mechanical [6] and corrosion [16] properties of BSS. The boron phase of cast BSS is generally distributed in the grain boundaries as a grid [17]. 

Rabino et al. investigated that aging studies of BSS, conducted at temperatures near the solidus, were used to determine the effects of elevated temperature exposures on impact properties and microstructure. These analyses indicated that the boride particle coarsening followed the theoretically predicted t^1/3^ time dependence and that the coarsening rate increased with increasing volume fraction of the boride phase [18]. Won et al. investigated the Cr and Ni contents of the austenite matrix: first, they were homogenized. Then, borated stainless steel was fabricated through a conventional ingot metallurgy-hot working process and annealed at 1180 °C for 192 h. Afterwards, the plate-like shape of (Fe,Cr)_2_B turned into a spherical shape [3].

In this study, the microstructure, distribution of boron chromium and iron, and its influence on the mechanical properties of BSS prepared by hot isostatic pressing (HIP) after heat treatment from 900 to 1200 °C were analyzed by field-emission scanning electron microscopy (FE-SEM).

## 2. Materials and Methods

The chemical composition of BSS prepared by PM was in accordance with Standard 304B7 of ASTM A887. Fe–B, Fe–Ni, Fe–Cr, and iron alloy were melted in a furnace and then atomized in a protective argon atmosphere. A spherical powder with the chemical composition shown in Table 1 was formed (Figure 1). The powder passed through a 74 μm sieve.

The powder was placed in a stainless-steel capsule, degassed to 10^−3^ Pa, heated to 500 °C, and sealed. The capsule was placed in an HIP furnace to raise the temperature and pressure in order to obtain a dense HIP billet. After the capsule was removed, one HIP billet was heat treated (HT). A chamber furnace was heated, the billet was held at different temperatures of 900 °C, 1000 °C, 1050 °C, 1100 °C, 1150 °C, and 1200 °C for 1 h, and then, it was water quenched.

Coupons of different temperature heat treatment were evaluated using quantitative X-ray diffraction (XRD) phase analysis according to Chinese Standard YB/T 5320. Microstructures were observed by FE-SEM with secondary electron (SE) and energy-dispersive spectroscopic (EDS) analysis. The polished and etched samples used for microstructure analysis were 20 mm × 20 mm × 10 mm. Distributions and contents of B, Fe, Cr and other elements at different locations were analyzed by FE-SEM-EDS using plane, line, and point scans. Strength and plasticity were tested according to standard ISO6892-1.

## 3. Results

### 3.1. Quantitative X-ray Diffraction Phase Analysis

Figure 2 shows the XRD patterns of the alloy after different temperature heat treatment. The prepared alloy comprised two main phases: the cubic austenite phase and the orthorhombic Fe_1.1_Cr_0.9_B_0.9_ boron phase. The “k-value” method was used to analyze the XRD patterns at different temperatures ranging from 900 to 1200 °C. The proportions of the two phases at diffident temperature were basically the same, the cubic austenite phase was approximately 90%, and the orthorhombic Fe_1.1_Cr_0.9_B_0.9_ boron phase comprised approximately 10%, indicating that temperature did not result in the boron phase change. 

### 3.2. Metallographic Structure

The corresponding SEM characterizations are shown in Figure 3a–f and Table 2. The grain size of boron phase after heat treatment at 900 °C (Figure 3a) was approximately 1 μm, and some grains already coalesced and grew up. The grain size of boron phase at 1000 °C (Figure 3b) was obviously larger than that at 900 °C; reaching 3–5 μm, there were many tens of nanometers of small grains around the large grain. The grain size at 1050 °C (Figure 3c) was similar to that at 1000 °C, and there were no small grains near the large grain, which indicates the dissolution of small grains and the re-deposition of the dissolved species on the surfaces of larger grains. The grain size at 1100 °C (Figure 3d) was slightly smaller than that at 1000 °C and 1050 °C, and the grain boundary was obvious. When the solution temperature increased to 1150 °C (Figure 3e), the grain size was larger than that at 1100 °C. At 1200 °C (Figure 3f), the grain size demonstrated obvious change, but the boundary between the boron phase and matrix was not obvious.

Figure 4a–f show the boron plane-scan distributions of BSS after heat treatment at different temperatures. Boron was mainly distributed in the boron phase and marginally in the austenite matrix. There was a small amount of boron in the austenite, as is shown in Figure 4, but no boron was found by point scanning in Figure 3. Possible reasons are that the accuracy of scanning by EDS is not very high, or boron phase grains with other orientations account for a small proportion in this interface and are not observed in the image. Boron distributions were concentrated in the boron phase at 900–1150 °C. The boron distribution was relatively dispersed at 1200 °C.

Table 2 summarizes the different element contents of SEM-EDS point scan analysis shown in Figure 3. Table 2 shows that the content of Cr in the boron phase was higher than that in the matrix, while Fe and Ni were opposite. EDS is not accurate for boron with a low atomic number, so boron was not detected at some points in the boron phase particles in Table 2. For the same reason, the detected boron element was also uneven. In addition, boron affected the proportion of other elements, so the data of Cr, Fe and Ni in boron phase grains were quite different. In order to detect the distribution of Cr, Fe, and Ni elements, the boron phase grain of the alloy after heat treatment at 1100 °C was analyzed by SEM-EDS line scan. Basically, contents of Cr and Fe were evenly distributed, although they fluctuated, as shown in Figure 5. 

### 3.3. Mechanical Properties

Table 3 shows the mechanical properties of BSS after heat treatment at different temperatures. Under 900–1100 °C heat treatment, the tensile strength and yield loudness were similar. When the heat treatment was 1150 °C, the tensile strength decreased by 59 MPa and the yield strength decreased by 54 MPa. The elongation and reduction of area increased continuously from 900 to 1100 °C and decreased slightly at 1200 °C. Under 1100 °C heat treatment, the alloy had excellent strength and plasticity.

Figure 6a–h show the fracture morphology of different temperatures. The fracture morphology at 900 °C shows that there were spherical particles, which separated from the matrix obviously. The separation of the particles from the matrix was obvious, which should be the crack source of fracture, resulting in the lowest plasticity. At 1000 °C, there were small fracture grains near the large fracture grains, which corresponded to more small grains near the large grains in Figure 3b. The fracture surface at 1050 °C was similar to that at 1100 °C, and the size of fracture grain was several microns. The transgranular fractures larger than 10 µm and large-sized pits were not found at 1100 °C, as shown in Figure 6d. The transgranular fracture larger than 10 μm of 1150 °C was found, as shown in Figure 6f. Large pits of about 30 microns, whose size was significantly larger than that of 1150 °C, were found in the fracture surface at 1200 °C (Figure 6f), which shows the decrease of strength. The number of pits at 1200 degrees was significantly higher than that of 1150 °C, as shown in Figure 6h.

## 4. Discussion

The solubility of boron in ferrite and austenite is very low, at only 0.001–0.0025% below 900 °C. A small amount of boron can improve the hardenability of steel. With an increase of temperature to 1072 °C, the solubility of boron in austenite is 0.016% [8]. During the preparation of boron steel powder, boron dissolves in the steel solution, and atomization involves cooling at high speed so boron has no time to diffuse, forming a supersaturated solid solution. In the densification process, the diffusion rate of boron increases with increase of temperature. Boron segregation to austenite grain boundaries is considered to occur by a non-equilibrium segregation mechanism [10] and is mainly concentrated on the grain boundary, filled with defects, and forms boron phases with Fe, Cr, or Ti. The interaction between dissimilar atoms becomes stronger in the order Co–B < Ni–B < Mn–B < Fe–B < Cr–B [19].

In the heat treatment process, boron diffuses to grain boundaries or defects in grains because the boron diffusion coefficient [8] is larger than that of iron and chromium in austenite [20]. The binding force of boron with iron and chromium is greater than that of other elements [19], so chromium and iron diffuse to the enrichment area of boron, and the content of chromium and iron in the boron phase is different from that of the austenite matrix phase. 

In the heat treatment process, both the nucleation and grain growth of the boron phase change with the temperature. The nucleation rate of the boron phase increases at 1000 °C, and the driving force of grain growth increases, too. The boron phase formed at the grain boundary merges and grows. Therefore, the grain size in Figure 3b is larger than that in Figure 3a, and there are many small grains near the large grains. During the growth of the boron phase at the grain boundary, chromium and iron also diffuse because of the bonding force with boron. With temperature exceeding 1050 °C, the small particles dissolute and re-deposit on the surfaces of large particles. The iron and chromium that diffuse into the vicinity of the boron phase form not only the boron phase (Fe_1.1_Cr_0.9_B_0.9_) with boron but also a solid solution with some of the nickel. The boron-containing phase particles form a eutectic-like structure of the boron phase (Fe_1.1_Cr_0.9_B_0.9_) and the CrFeNi solid solution. As the solid solution phase is more stable, the boron phase inside the grain transforms into the solid solution phase. Boron has limited solubility in the solid solution phase and continues to diffuse to new grain boundaries, but due to diffusion and phase transition time limitations, boron remains partially inside the grain. With temperature exceeding 1150 °C, the unbalanced grain phase of local austenite matrix results in abnormal grain growth due to element diffusion, as shown in Figure 6f. With temperature exceeding 1200 °C, the grain size of abnormal growth is larger than that at 1150 °C, because boride melts at this temperature and element diffusion accelerates. At 1200 °C, the diffusion rate of the elements is accelerated, and the boron element is more likely to be segregated at the grain boundaries to form the boron phase (Fe_1.1_Cr_0.9_B_0.9_), so the boron-containing phase in Figure 3f is not obvious from the matrix phase. The melted boron phase (Fe_1.1_Cr_0.9_B_0.9_) is distributed around the large grains, which is the reason why there are more large-sized pits in the fracture morphology, as shown in Figure 6h. 

The strength and toughness of the alloy were attributed to the effect of different microstructures caused by different heat treatment temperatures. The boron phase precipitating at the temperature of 900 °C plays a strengthening role, but the plasticity is low due to the small grain size of boron phase and weak bonding force. With temperature exceeding 1000 °C, the strength does not decrease, while the plasticity increases due to the growth of boron phase. With temperature exceeding 1150 °C, the strength decreases due to the abnormal grain growth. With temperature exceeding 1200 °C, the abnormal growth is more obvious, and both the strength and the plasticity drop significantly.

## 5. Conclusions

Borated stainless steel prepared by PM consists of two phases: an austenite matrix and a hard boron phase (Fe_1.1_Cr_0.9_B_0.9_) distributed in chromium-rich austenite after heat treatment at different temperatures.Due to the diffusion coefficient and binding force between the elements varying at different temperatures; it leads to a different distribution of each element in the boron phase and austenite matrix.Alloy with good strength and plasticity can be obtained when the heat treatment temperature of high boron stainless steel reaches 1000–1150 °C. The tensile strength is approximately 800 MPa, and the elongation is approximately 20%.

## Figures and Tables

**Figure 1 materials-14-04646-f001:**
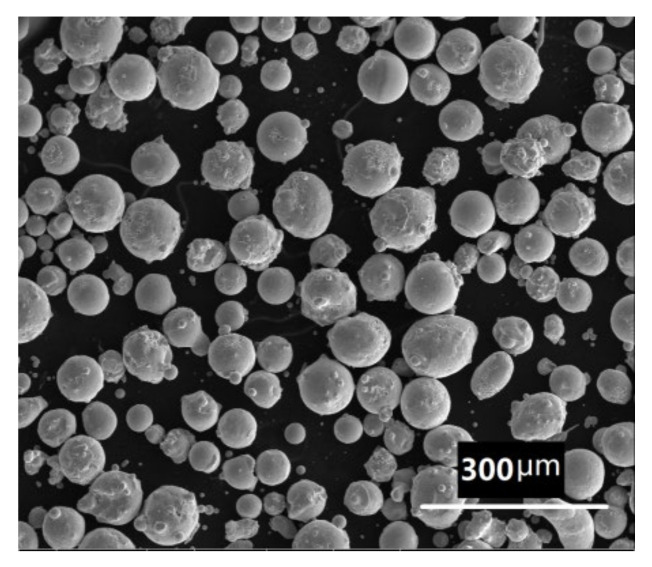
Metallographic of BSS powder.

**Figure 2 materials-14-04646-f002:**
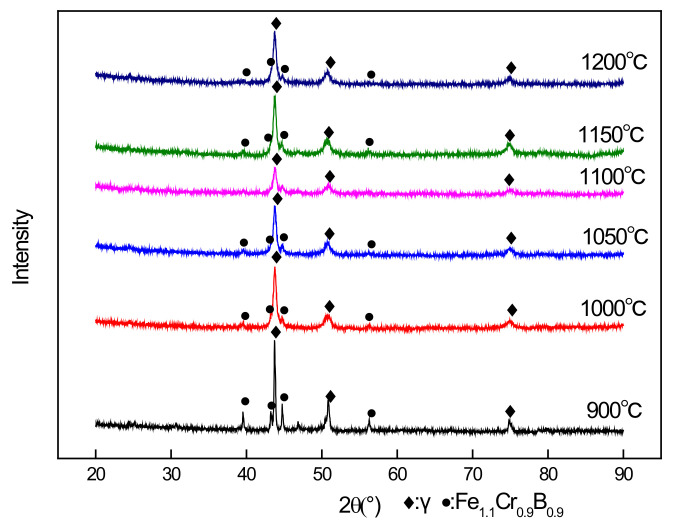
XRD of BSS after different temperatures.

**Figure 3 materials-14-04646-f003:**
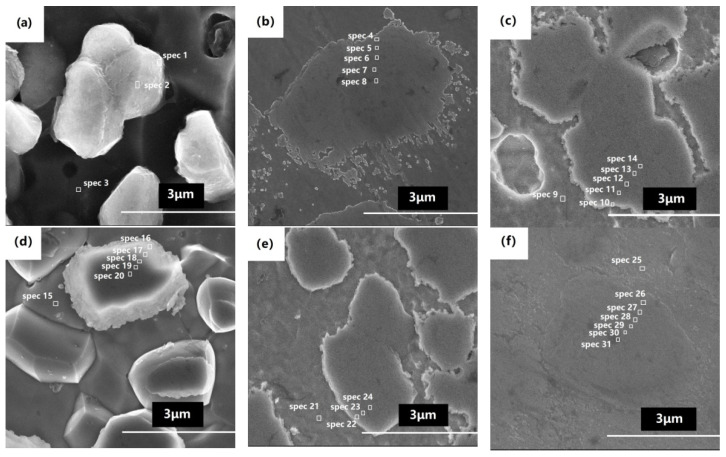
**Microstructure and** SEM-EDS point analysis of BSSs after heat treatment at (**a**) 900 °C, (**b**) 1000 °C, (**c**) 1050 °C, (**d**) 1100 °C, (**e**) 1150 °C and (**f**) 1200 °C.

**Figure 4 materials-14-04646-f004:**
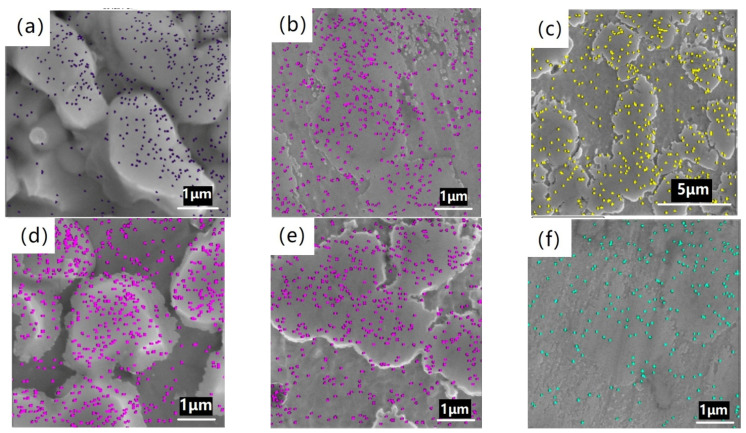
SEM-EDS plane analysis of boron after heat treatment at (**a**) 900 °C, (**b**) 1000 °C, (**c**) 1050 °C, (**d**) 1100 °C, (**e**) 1150 °C and (**f**) 1200 °C.

**Figure 5 materials-14-04646-f005:**
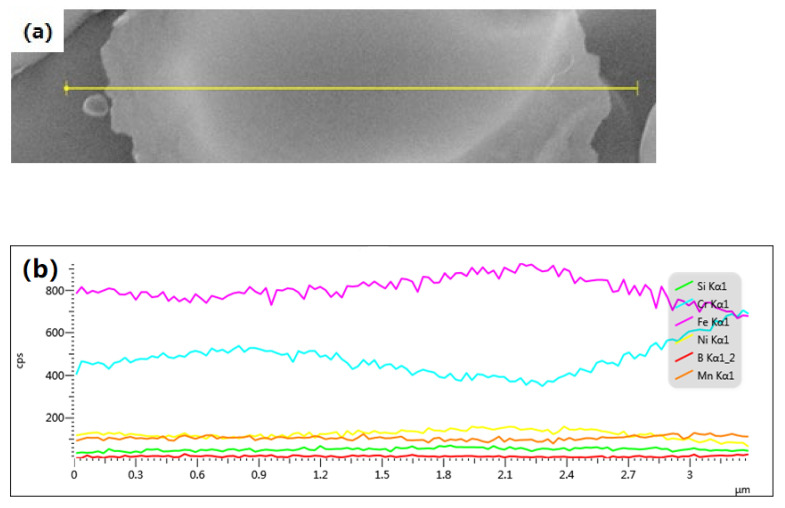
SEM-EDS line analysis of boron after heat treatment at 1100 °C (**a**) line scan across grains; (**b**) curve of the relative content for different elements of the line scan.

**Figure 6 materials-14-04646-f006:**
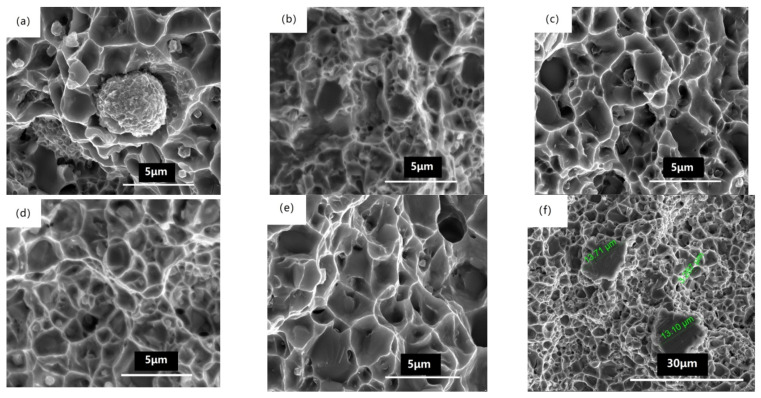
The fracture morphology of BSSs after heat treatment at (**a**) 900 °C, (**b**) 1000 °C, (**c**) 1050 °C, (**d**) 1100 °C, (**e**) 1150 °C, (**f**) 1150 °C, (**g**) 1200°C and (**h**) 1200°C of tensile tests.

**Table 1 materials-14-04646-t001:** Chemical composition of investigated alloys in mass%.

Element	B	C	Cr	Ni	Mn	Si	Ti	P	S
mass%	1.86	0.020	19.30	14.10	2.00	0.66	—	0.0070	0.0051

**Table 2 materials-14-04646-t002:** Elements content of SEM-EDS point scan analysis of BSS.

	Spectrum	B	Fe	Cr	Ni	Mn	Si
Figure 3a	1	-	62.6	21.9	12.5	2.2	0.8
	2	-	46.9	50.1	3.0	-	-
	3	-	64.8	16.5	15.9	1.9	0.8
Figure 3b	4	-	65.4	21.6	12.6	-	0.4
	5	23.2	42.0	27.1	6.2	-	-
	6	17.6	40.6	35.3	4.5	-	-
	7	23.2	33.7	38.8	2.3	-	-
	8	27.4	30.2	38.9	1.7	-	-
Figure 3c	9	-	68.5	13.2	15.3	1.9	1.0
	10	5.7	60.4	16.6	14.2	2.1	0.3
	11	5.3	56.7	24.4	11.1	1.9	0.6
	12	5.8	48.7	38.8	4.7	2.1	-
	13	-	48.6	44.6	4.1	2.3	0.5
	14	-	50.3	45.7	4.0	-	-
Figure 3d	15	-	66.7	14.0	16.5	1.9	0.9
	16	17.7	49.6	20.4	9.8	1.8	0.8
	17	17.7	49.6	21.0	8.9	2.0	0.8
	18	15.5	49.8	22.3	9.6	1.9	0.9
	19	18.4	48.9	21.2	9.0	1.7	0.7
	20	17.4	47.0	27.9	7.2	-	0.5
Figure 3e	21		64.5	16.7	15.5	2.5	0.8
	22	-	65.3	16.0	15.7	1.8	1.2
	23	-	64.2	18.3	14.3	2.3	0.9
	24		60.8	24.5	12.1	1.9	0.7
Figure 3f	25	-	65.6	16.5	14.9	2.2	0.8
	26	11.8	47.2	33.0	5.6	2.1	0.5
	27	-	48.4	46.2	2.9	2.5	-
	28	-	45.3	49.8	2.3	2.5	-
	29	8.3	40.2	49.7	1.7	-	-
	30	-	43.4	52.7	1.2	2.7	-
	31	6.2	41.4	51.2	1.2	-	-

**Table 3 materials-14-04646-t003:** Mechanical properties.

Heat Temperature (°C)	Tensile Strength (MPa)	Yield Strength (MPa)	Elongation (%)	Reduction of Area (%)
900	804	398	18	14
1000	802	396	20	18
1050	798	390	20	20
1100	792	406	23	22
1150	776	380	20	23
1200	745	352	21	20

## Data Availability

Data available in a publicly accessible repository.

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
