# Peer review of "Effect of Heat Treatment Temperature on Microstructure and Properties of PM Borated Stainless Steel Prepared by Hot Isostatic Pressing"

_materials, 2021, doi:10.3390/ma14164646_

Round 1
Reviewer 1 Report
This paper contain the interesting experiments about the effect of heat treatment temperature for borated stainless steel.
However, there are some concerns about the evaluation and interpretation of the experimental results, which may require significant improvement.
Specific comments.
Fe1.1Cr0.9B0.9
What do you think the chemical formula of the boron phase represents? If the composition changes significantly from the analysis results, the chemical formula does not apply.
As for Figure 2.
Please describe the results regarding the relationship between heat treatment temperatures and comprised two phases.
As for Figure 3 and Table 2.
Where does the boron phase indicate in Figure 3?
It is strange that the place where the boron are detected or not detected exist in the same grain indicated in Table 2.
Isn't it possible to consider the drift of the sample during analysis?
Results of Figure 4.
Does it indicate that the distribution of boron has not been detected?
As for Figure 5.
The concentration distribution is intense, and it seems that it is not solid solution or compound.
Figure6
The resulting figure is not shown.
L202 10-10cm2/s → 10-10cm2/s
Author Response
Dear reviewer:
Thank you very much for your attention, evaluation and comments on our paper. We have revised the manuscript according to your kind advice and detailed suggestions. Please see the attachment.
Best regards
Sincerely yours

Reviewer 2 Report
Authors studied the effect of heat-treatment on the microstructure and mechanical properties of a Borated stainless steel. BSS was prepared by PM process and isostatic pressing. The microstructure was studied by SEM and XRD. The manuscript needs major revision as in contains various typs and grammatical errors. Following are the specific comments
Abstract line 16 is incomplete.
Authors need to stick to either at% or mass%. The reviewer recommends at%.
Authors need to provide the which SEM mode imaging (SE or BSE) was performed.
Figures 1, 3 and 4 need better scale bars.
There are symbols in Figure 3, as the journal accepts articles in English, please remove these symbols.
Authors have used very confusing language while explaining Figure 4 by referring it as plane-scan view, point scan and plant scan at various locations.
Figure 6 is missing in the manuscript
Author Response
Dear reviewer:
Thank you very much for your attention and the evaluation and comments on our paper. We have revised the manuscript according to your kind advices and detailed suggestions. Please see the attachment.
Best regards
Sincerely yours
Yanbin Pei

Round 2
Reviewer 1 Report
This paper contain the interesting experiments about the effect of heat treatment temperature for borated stainless steel.
Please reconsider or add explanations regarding the following viewpoints.
As for Figure 3 and Table2.
Does Ni not contain in the boron phase Fe1.1Cr0.9B0.9? Or does Ni and other elements also contain in the boron phase?
If the composition close to Fe: Cr = 1.1: 0.9 is considered to be the boron phase, please indicate which spectrums of Table 2 and region in Figure 3 are considered to be the boron phase.
As For Figure 2 and Figure 4.
It is described that the proportion of boron phase does not change in the XRD of Figure 2 even if the boron distribution disappears in Figure 4 at 1200 ℃.
Please add a description on this point.
As for Figure 6.
L206. 1200 ℃ (Fig.6 (f)) → 1200 ℃ (Fig.6 (g))
Figure 6 shows a reduced view of the fracture surface of (f) and (g), showing the correlation between the large pits and its mechanical properties.
The number and proportion of large pits is almost non-existent up to 1100 ℃, and is it higher at 1200 ℃ than at 1150 ℃?
Please add a description on this point.
Author Response
Dear Reviewer:
Thank you very much for your attention and comments on our paper. We have revised the manuscript according to your kind advices and detailed suggestions.
Best regards
Sincerely yours
Yanbin Pei

Reviewer 2 Report
Please provide the unit of the composition of B in the steel.
Author Response

(The authors gave the same response as above.)
